# Concentration and Spatial Distribution of Potentially Toxic Elements in Surface Soil of a Peak-Cluster Depression, Babao Town, Yunnan Province, China

**DOI:** 10.3390/ijerph18063122

**Published:** 2021-03-18

**Authors:** Hongyu Tian, Cheng Zhang, Shihua Qi, Xiangsheng Kong, Xiangfei Yue

**Affiliations:** 1School of Environmental Studies, China University of Geosciences, Wuhan 430078, China; tianhongyu@cug.edu.cn (H.T.); shihuaqi@cug.edu.cn (S.Q.); 2Key Laboratory of Karst Dynamics Ministry of Land and Resources Guangxi, Institute of Karst Geology, Chinese Academy of Geological Sciences, Guilin 541004, China; zhangcheng@mail.cgs.gov.cn (C.Z.); yuexf06@126.com (X.Y.)

**Keywords:** potentially toxic elements, surface soil, peak-cluster depression, spatial distribution, Karst region

## Abstract

Potentially toxic elements (PTEs) in Chinese agricultural soils, including those in some heritage protection zones, are serious and threaten food safety. Many scientists think that these PTEs may come from parent rock. Hence, at a karst rice-growing agricultural heritage area, Babao town, Guangnan County, Yunnan Province, China, the concentrations of eight PTEs (As, Cd, Cr, Cu, Hg, Ni, Pb, and Zn) were determined in 148 surface soil, 25 rock, and 52 rice grain samples. A principal component analysis (PCA) and hierarchical cluster analysis were used to divide the surface soil into groups, and inverse distance weighting (IDW) was used to analyze the spatial distribution of PTEs. Soil pollution was assessed with the geoaccumulation index (Igeo). The results show that Cd, Hg, Zn, and Cr were polluting the soil (average Igeo > 0). The highest concentration of PTEs was distributed in the southwest of Babao town in the carbonate rock area, which had the highest pH and soil total organic carbon (Corg), Mn, and TFe_2_O_3_ contents. PCA biplots of soil samples showed that the carbonate rock area was associated with the most species of PTEs in the study area including Pb, Cd, Hg, As, and Zn. The clastic rock area was associated with Cu and Ni, and the lime and cement plants were associated with CaO, pH, Corg, TC, and aggravated PTE pollution around factories. In high-level PTE areas, rice was planted. Two out of 52 rice grain samples contained Cd and 4 out of 52 rice grain samples had Cr concentrations above the Chinese food safety standard pollutant limit (Cd 0.2 mg/kg; Cr 1 mg/kg). Therefore, the PTEs from parent rocks are already threatening rice safety. The government should therefore plan rice cultivation areas accordingly.

## 1. Introduction

Cr, Ni, Cu, Zn, Cd, Pb, Hg, and As are the potentially toxic elements (PTEs) with the highest levels of toxicity in the environment [1]. They are toxic, persistent in the environment, and have a bioaccumulative nature, making them threats to the environment and human health [2]. Human activities and natural processes (such as rock weathering [3] and volcano eruptions [4]) are two sources of PTEs in the environment. Previous studies have shown that the main reason for a high PTE content in soil is the weathering of parent rock [5,6]. PTEs are not only inherited from parent rock but also enriched or leached by weathering process differences [7,8]. During the weathering process, limestone regions appear to be more conducive to the surface aggregation of elements than sandstone regions [7]. Hence, carbonate rock areas can become natural PTE geochemical abnormal areas.

Southwest China is the largest karst region in China, and it has a wide range of carbonate rocks. In southwest China, the background concentrations of some PTEs, such as Cr (133.3 mg/kg), Cd (1.2 mg/kg), As (28.5 mg/kg), Zn (135.9 mg/kg), and Ni (66.7 mg/kg) in Guizhou province and Cu (48.2 mg/kg) in Yunnan province [9], are higher than in other regions. Pesticide and fertilizer application [10], mine exploitation [11], and wastewater discharge [12] increase the concentration of PTEs in the soil. A national survey found that PTE pollutants threaten 19% of China’s farmland soil, mainly in southwest China [13]. Even some critical, protected rice-growing areas, such as the Hani terrace wetlands in Yunnan Province, have large amounts of Cd in the soil [14]. As edible parts of crops grown in soil contaminated with PTEs can accumulate PTEs, endangering humans system through their consumption in foods [15,16], PTE pollution of agricultural soil in the karst area in Southwest China has attracted widespread concern from many scientists.

Babao town is one of the vital rice planting conservation areas in Yunnan Province, Southwest China and is one of world cultural heritage sites of United Nations Educational, Scientific, and Cultural Organization (UNESCO). The town is located in a typical karst zone and is famous for its high-quality rice. Historically, the rice was used as a tribute. Now, the rice has become a geographical indication product [17]. Due to topographical limitations, agricultural production has become the main contributor to the local economy. Studies have shown that there is serious Cd and Hg pollution in the soil around this area [18,19]. However, little information is available on the quality of the overall soil–rice system in Babao town. Pollution may derive from parent materials and threaten the safety of the Babao production area. Therefore, we collected 148 surface soil samples, 52 rice grain samples, and 25 rock samples from the town to investigate the soil and rice situation. Cluster analysis was used to classify soil samples, and the results were combined with a geological statistical analysis to study the soil spatial variation of PTEs and the origins and associated risks of PTEs.

## 2. Materials and Methods

### 2.1. Study Area

The study area was located in Babao town, Guangnan County, Wenshan Zhuang-Miao Autonomous Prefecture of Yunnan Province, China (105°12′–105°28′ E, 23°41′20′′;–23°46′60′′; N) (Figure 1). The annual average temperature in this area is 16.7 °C (based on the Guangnan county data from 1971–2000, http://www.weather.com.cn/cityintro/101290607.shtml accessed on 11 March 2021), and the average annual rainfall is 1104.65 mm (based on Funing station observations from 2001–2015): ideal climate conditions for rice cultivation. The farmland has a fertile texture with limestone soils that are fit for rice growth. Hence, Babao town has a history of planting rice of more than 700 years.

Babao town is a typical peak-cluster depression area with elevation ranging from 1100 to 1300 m, and rice planting is mainly concentrated in three areas (Figure 1a–c). The three main parts contain different types of rock. One part contains carbonate rocks, with limestone and dolomite as parent rocks; the second part contains mixed rocks with limestone, dolomite, and clastic rock; and the third part contains clastic rocks. Hence, there are also three different soil types: yellow-red soil, red soil, and brown soil. In the Babao town center, there is a lime plant and a cement plant. Around the town, there are many private factories that mine titanium, limestone, shale, and titanite.

### 2.2. Sample Collection and Chemical Analysis

The following sampling and analysis methods were used: “Specification of Multi-Purpose Regional Geochemical Survey (1:250,000) (DZ/T 0258-2014)”, “Analysis Methods for Regional Geochemical Sample (DZ/T 0279-2016)” issued by Ministry of Land and Resources of the People’s Republic of China, and “Technical requirements for analysis of samples for ecological geochemical evaluation (DD2005-03)” issued by the China Geological Survey. These articles also adopted the sampling and testing methods promulgated by the China Geological Survey and Ministry of Land and Resources of the People’s Republic of China [19,20]. The study used 1000–1500 g samples of the following types: 148 surface soil (0–20 cm) samples, 52 rice grain samples, and 25 rock samples. The locations of the sampling points were accurately recorded using a GPS instrument. Rock samples were collected in outcrop, including 15 limestones, 2 siliceous rocks, 2 dolomites, 1 sandstone, 3 mudstones, 1 mudstone interbedded with limestone and 1 limestone interbedded with mudstone. A regular 1 × 1 km uniform grid sampling method was adopted to sample the surface soil. Then, the four 1 × 1 km subsamples were blended into one sample and air-dried at room temperature. After the soil was dried entirely, the soil was crushed with a wooden stick, debris such as gravel and plant residues was removed from the samples, the soil was screened with 200 mesh (0.074 mm) nylon, and samples of 500 g were collected with the quartering method and put into polyethylene plastic bottles. All rock samples were also ground and then sieved to get powders (<0.074 mm) for further chemical analysis. The rice grains were sampled from 10–20 rice in a 0.1 hm^2^ sampling unit. Samples of rice were then dried and 500 g rice grain samples were placed into polyethylene plastic bottles for sample delivery.

An elemental analysis of samples was carried out by the Guangdong provincial geological experimental testing center and the Hubei provincial geological experimental testing center. Before testing, the rice grains were hulled, crushed, sifted through 40–60 mesh screens. The method used to determine the total PTE content in the soil, rice grain, and rock samples was as follows: Samples (0.3 g each, approximately) were digested in HNO_3_–HCl–HClO_4_–HF acid mixture in a Teflon digestion vial and then heated. They were then extracted with hydrochloric acid and diluted to a certain volume. Then, the Cd, Pb, Zn, Cu, and Mn contents were determined (ICP-MS, Nexion 300X, Perkin Elmer Company, Waltham, MA, USA). The As, Cr, and Hg contents were measured using atomic fluorescence spectrometry (AFS-820, Jitian Instrument Company, Beijing, China). The Ni, CaO, MgO, and TFe_2_O_3_ contents were measured with an X-ray fluorescence spectrometer (XRF-1800, Shimadzu, Kyoto, Japan). The soil pH value was determined with a pH meter, and the soil–water ratio was 2.5:1. The soil total organic carbon (Corg) was determinated by titration with burette and total carbon (TC) contents were determined with a High frequency infrared carbon sulfur analyzer, Model CS-206 (Shanghai Baoying Technology Company, Shanghai, China).

To ensure the accuracy of the sample analysis, 5% of the samples were randomly selected for parallel testing, and the repeat test pass rate was 100%. Reagent blanks, standard reference soil/rock samples (GBW07443), and standard rice samples (GBW10010) were obtained from the Center of Standard Materials of China to monitor the determination quality. The recovery of samples spiked with standards ranged from 89.8% to 107.9%. The LOD (limit of detection) ranged from 0.0005 mg/kg (Hg of soil sample) to 3 mg/kg (Cr of soil sample). The Cu and Pb contents were below the detection limit (Cu 1 mg/kg, Pb 0.05 mg/kg) in only a few rice grain samples. The values for these samples were set to half the detection limit.

### 2.3. Principal Component Analysis (PCA)

The principal component analysis (PCA) is a multivariate statistical method that uses the idea of dimensionality reduction and divides multiple indicators into several components on the premise of losing little information. The Kolmogorov–Smirnov test was used to assess the normality of the data set distribution. The results show that most elements did not have a normal distribution. PCA was performed before the data was log-ratio transformed (clr), and the numerical output with the eigenvalues was ≥1.

### 2.4. Hierarchical Cluster Analysis and Box Plot

The hierarchical cluster analysis was conducted with Origin 2019, and a z-score was used for the standardized data analysis. The proposed approach is based on sampling sites observed using the Ward Method with the Euclidean distance and the determination of clustroids by the sum of distances. Similar sampling points belong to the same cluster and are different from data within other clusters [21]. The selection of the number of clusters was based on a visual analysis of the dendrogram and was supported with the principal component analysis. Then, box plots and geostatistical analysis were used to reveal the spatial distribution characteristics of PTEs. Box plots were used to describe the distribution of elements in each cluster with the median, mean, upper and lower extremes of the range of values, and the 75 and 25 percentile values [22].

### 2.5. Spatial Distribution Analysis of Elements

The inverse distance weighting (IDW) method was used in this study to analyze the distribution of PTEs. Both IDW and kriging interpolation methods can provide high accuracy for predicting the average concentrations of elements in the soil. Each of them has its shortcomings. IDW enlarges the estimated pollution area [23]. The kriging method has a smoothing effect, which makes it difficult to fully identify highly polluted areas [24]. However, IDW is simpler than kriging and does not require a strictly normal data distribution. Therefore, IDW was carried out with ArcGIS V.10.6 software (Esri, Redlands, CA, USA) using the geostatistical analysis module.

### 2.6. Method of Assessing Soil Pollution

The geoaccumulation index (Igeo) takes the influences of human activities and natural geological processes of background values into account and can quantitatively assess the extent of PTE contamination of sediments or other materials [25]. Therefore, the Igeo was used to assess surface soil pollution. The Igeo was calculated as follows (Equation (1)):(1)$$I_{\mathrm{geo}}=\log_{2}\left(\frac{C_{i}}{1.5B_{i}}\right)$$
where *C_i_* is the soil content of the element, *i*, and *B_i_* is the geochemical background value in the soil. The background values used in the study were based on the soil background values in Yunnan Province [26]. The classifications of PTE pollution were as follows: *I*_geo_ ≤ 0, practically unpolluted; 0 < *I*_geo_ ≤ 1, slightly polluted; 1 < *I*_geo_ ≤ 2, moderately polluted; 2 < *I*_geo_ ≤ 3, heavily polluted; *I*_geo_ > 3, extremely polluted [25].

### 2.7. Statistical Analysis

Sampling sites were mapped using ArcGIS 10.6 (Esri, Redlands, CA, USA). The data were processed in Excel and SPSS v22 (IBM, Armonk, NY, USA). Charts, and graphs of data were produced using Origin 2019 (OriginLab, Northampton, MA, USA).

## 3. Results and Discussion

### 3.1. Descriptive Statistics of Soils, Rocks, and Rice Grains

The descriptive statistics of soil characteristics in Babao Town are shown in Table 1. The pH of the study area ranged from 5.19 to 8.07, and 56.8% of soil had a pH of above 7. Most of the soil was neutral or alkaline. Except for Cu, the variable coefficients (CV) of PTEs in the soil were larger than 35%, reflecting the highly variable PTE distribution in the soil of the study area [27], The concentrations (mg/kg) of PTEs in the soil were in the following decreasing order: Zn > Cr > Ni > Cu > Pb > As > Hg > Cd (Table 1).

The average values of all PTEs were higher than the background values for Yunnan, which indicated this area is enriched with PTEs [26]. The average concentrations of As, Cd, Cr, Cu, Hg, Ni, Pb, and Zn were 2.09, 27.33,1.73, 1.87, 3.78, 2.26, 1.42, and 2.93 times higher than the mean values found in the first national soil pollution survey (As 12.1 mg/kg, Cd 0.225 mg/kg, Cr 68.5 mg/kg, Cu 27.1 mg/kg, Hg 0.087 mg/kg, Ni 29.6 mg/kg, Pb 31.2 mg/kg, Zn 79.0 mg/kg) conducted in China, respectively [9]. In particular, the Cd concentration was much higher than that in other karst areas in China, such as Quanzhou County (Cd mean value 0.43 mg/kg) [25] and Heng County (Cd mean value 0.60 mg/kg) in Guangxi [29].

The surface soil was divided into groups by the hierarchical cluster analysis. Figure 2 shows a dendrogram of the cluster analysis of the selected variables. For this study, a dissimilarity value of approximately 40 was selected to split the dendrogram into four clusters. It can be seen from Figure 1 and Figure 3 that the points of cluster 1 were located across all the soil types and were mainly in the mixed area of yellow-red soil. Points of cluster 2 were in the clastic rock area with red soil, and cluster 3 was in the carbonate rock area with brown soil. We found points of cluster 4 around the lime plant and the cement plant (Figure 3). Hence, cluster 1, cluster 2, cluster 3, and cluster 4 represent soil samples affect-ed by both two types of rocks, clastic rock, carbonate rock, and human activity, respective-ly. 

The numerical output with eigenvalues ≥1 from the principal component of soil are listed in Table 2. PCA summarizes all data in three principal components (PC), representing 77.93% of the data total variance (PC1: 49.28% of the variance, PC2: 18.29% of the variance, and PC3: 10.36% of variance). With the aid of Table 2 and Figure 4, the following observations were made.

PC1 has positive contributions from all elements. PC2 has positive contributions from CaO, pH, Corg, TC, Pb, Cd, and Hg, and negative contributions from MgO, TFe_2_O_3_, Cu, Ni, Mn, and As. PC3 has positive contributions from MgO, TFe_2_O_3_, Cu, Ni, and Cr, and negative contributions from Zn, Mn, and As. The surface soil samples of cluster 2 (surface soil samples affected by clastic rock) were associated with MgO, TFe_2_O_3_, Cu, and Ni and occupied the +PC3 quadrant (Figure 4). The surface soil samples of cluster 3 (surface soil samples affected by carbonate rocks) were associated with Pb, Cd, Hg, As, Zn, and Mn and occupied the −PC3/±PC2 quadrant. The surface soil samples of cluster 4 (surface soil samples affected by human activity) were associated with CaO, pH, Corg, and TC occupied the +PC2 quadrant.

Within the study area, formations with carbonate rocks as the main rock type (P_2_, D_2_, D_3_, and C_2_) and those with mudstone and sandstone as the main rock types (Є_3_, O_1_, and D_1_) displayed different chemical characteristics. All mean PTE concentration values in clastic rocks were higher than those in carbonate rocks; however, the mean values of Cd in carbonate rocks and clastic rocks were almost equal, and the median value of Cd in carbonate rocks was higher than that in clastic rocks (Table 3). Generally, carbonate rocks contain lower concentrations of Cd [30]. In this area, the Cd content in carbonate rocks was higher than that in other areas, such as in Jishou of Huanan Province (0.05 mg/kg), Huaxi of Guizhou Province (0.38 mg/kg), and Longzhou of Guangxi (0.20 mg/kg) [25]. Forty-one percent of carbonate rock samples (R2, R3, R11, R13, R15, R16, R18) had Cd concentrations above 1 mg/kg, while only the concentration in R20 (clastic rock) was above 1 mg/kg (3.55 mg/kg). In addition, the limestone interbedded with mudstone (R24) in the mixed area had the highest concentration of Cd with 11.3 mg/kg.

The average concentrations of PTEs in rice grains were 0.12 mg/kg for As, 0.043 mg/kg for Cd, 0.40 mg/kg for Cr, 1.64 mg/kg for Cu, 0.004 mg/kg for Hg, 0.31 mg/kg for Ni, 0.067 mg/kg for Pb, and 17.92 mg/kg for Zn, respectively (Table 1). Except for As and Zn, the coefficients of variation (CVs) of PTEs in rice grains were higher than 35%, reflecting the strong variation in concentrations of PTEs in rice grains in the study area.

### 3.2. Spatial Distribution

The surface soil was divided into four clusters to display the spatial distribution. Cluster 1 had the lowest concentrations of Cr (38.40–136.50 mg/kg, median 75.50 mg/kg), Cu (14.73–82.79 mg/kg, median 41.08 mg/kg), Mn (204.16–3015.83 mg/kg, median 1229.44 mg/kg), Ni (20.62–96.30 mg/kg, median 43.19 mg/kg), Zn (52.40–268.00 mg/kg, median 109.40 mg/kg), TFe_2_O_3_ (2.23–9.69%, median 5.79%), MgO (0.25–2.91 mg/kg, median 0.77 mg/kg), and CaO (0.14–10.13%, median 0.51%) (Figure 3 and Figure 5). Cluster 2, the clastic rock area, had the lowest concentrations of As (8.14–24.06 mg/kg, median 13.06 mg/kg), Cd (0.18–3.48mg/kg, median 0.565 mg/kg), Hg (0.051–0.297mg/kg, median 0.088 mg/kg), and Pb (11.70–35.10 mg/kg, median 17.40 mg/kg). Similarly, the average TC, Corg, and pH values were also the lowest in cluster 2 (clastic rock area). Interestingly, the average concentrations of Cu, Ni, TFe_2_O_3_, and MgO in the clastic rock area were the highest (Figure 3 and Figure 5). Cu, Ni, and Fe are siderophile elements that generally exist in basic rock and ultrabasic rock. Previous studies have shown that areas with high concentrations of TFe_2_O_3_ generally have high PTE concentrations [30]. However, our result was opposite to theirs. The reason for this may be that the pH values of clastic rock areas are lower—below 8.0 (Figure 5). The non-specific adsorption capacity of iron oxides for cationic PTEs when soil pH values are below 8.0 is lower than that for soil with pH values above 8.0, and the balance between negative and positive charges is not generated in areas with low pH values [31].

Carbonate rock areas were mainly found in the southwest part of Babao town (Figure 3) and had the highest concentrations of PTEs, except for Cu and Ni (Figure 5). As shown in Figure 5, except for Cu and Ni, the median concentration of PTEs was much higher than that in other areas with a pH range of 5.81–8.03 and had the second-highest median of 7.33. In general, an increase in alkalinity decreases the mobility of PTEs in the soil [32]. Therefore, the concentration of PTEs is higher in carbonate rock area with a high pH.

The soil types present in carbonate rock area are brown soil and scattered brown silt/sand soil. This may be why Cluster 3 (soil samples affected by carbon rock) had the second-highest Corg and TC values (Figure 5). Soils in carbonate rock area can have high Corg contents [33]. Meanwhile, the carbonate rock areas had the highest Mn content and the second-highest TFe_2_O_3_ content in the soil (Figure 5). Hence, organic materials linked together with colloidal particles of Fe/Mn in the soil can chelate PTEs, which limits the dispersion of metallic elements [32]. Therefore, areas with high levels of PTEs were mainly distributed in carbonate rock zones.

The highest average CaO, TC, Corg, and pH zones were found in cluster 4. Apparently, the use of carbonate rocks by plants has resulted in elevated levels of calcium in the soil. The pH of the soil is mainly controlled by calcium [29]. A high calcium concentration results in a high pH and decreases the mobility of PTEs [8]. Besides this, lime and cement may be spilled during transportation, and transport itself can increase Cd and Pb levels [34,35]. Therefore, it was not surprising that areas with significantly high Cd, Cr, Pb, and Zn concentrations were found around the two factories, and the concentration of PTEs was only lower than that in carbonate rock areas (Figure 3 and Figure 5).

### 3.3. Enrichment and Depletion of PTEs in Soils

In addition to enrichment of the concentrations of PTEs (such as Cd) in the soil by lime and cement factories, atmospheric deposition and fertilizer application are important ways to increase the soil PTEs [36]. However, the proportion of PTEs transferred from the fertilizer to the soil is very low compared to other origins [25]. Hence, studies have proposed that PTEs in the soil are usually derived from parent rock [25,37].

The ratio of the mean rock and soil concentration (PTEs _soil_/PTEs _rock_) was calculated under the assumption that the soil in the investigated area was directly derived from the chemical succession of the bedrock [32]. PTEs _soil_/PTEs _carbonate rock_ ratio was calculated using sampling points of cluster 3, and PTEs _soil_/PTEs _clastic rock_ ratio was calculated using sampling points of cluster 2. A ratio of >1 indicates enriched soil. The ratios of PTEs soil/PTEs rock in both two types of rocks were above 1, indicating that soils generated by carbonate rock and clastic rock were both rich in PTEs.

The PTEs _soil_/PTEs _rock_ of carbonate rock was 2.3(Ni)-32.8(Zn) times as much as that of clastic rock (Table 2). The PTEs _soil_/PTEs _rock_ of carbonate rock ranged from 6.64 (Ni) to 83.53 (Zn), while in clastic rock, it ranged from 1.06 (Pb) to 3.30 (Cu). The reason for the higher PTEs _soil_/PTEs _rock_ ratio in carbonate rock area is because of the large rock/soil volume change that occurred during the process of weathering carbonate rock into the soil [38]. The PTEs that were not evacuated would be significantly enriched in the weathering residue [3]. Those PTEs would have undergone coprecipitation or adsorption by Fe oxy-hydroxides [33] and Mn oxy-hydroxides [39] in residual soil. Meanwhile, higher pH and Corg values can lead to increased adsorption or co-precipitation with oxides/hydroxides of cations of PTEs [40]. Soil in carbonate rock areas had high pH and Corg values and high Mn and TFe_2_O_3_ concentrations (Figure 3 and Figure 5), which are good conditions for PTE accumulation. Therefore, the PTEs from natural processes (parent rock) were most enriched in the carbonate rock areas.

In the carbonate rock area, Cd was most enriched in the soil. Xie et al. [18] found that excess Cd areas were mainly distributed in lime soil from P_1_y, Ch, Csd, D_3_w, D_2_d, Dd, and Dg formations in Nasa of Guangnan county (80 km to the west of Babao town). In this study, areas of excess Cd were mainly located in P_2_y and D_3_g (Figure 1 and Figure 3). The Cd _soil_/Cd _rock_ ratio in carbonate rock area was as high as 16.56, larger than ratios found in Lower Burgundy (ranged from 4.6 to 5.7) in France [41] and Huaxi of Guizhou Province (the ratio was 8.3) [25]. Carbonate rock had a higher Cd content (mean value 0.69 mg/kg, median value 0.40 mg/kg) (Table 3). Hence, rocks with high Cd concentrations cause the Cd _soil_ to be of a high level (Figure 3, Table 1), making Cd concentration as high as 26.70 mg/kg in the study area, much higher than that in other carbonate areas.

### 3.4. Risk Evaluation

The Igeo of PTEs was obtained using formula 1. Igeo spatial distributions of elements are shown in Figure 6. The Igeo values of As, Cd, Cr, Cu, Hg, Ni, Pb, and Zn ranged from −2.26 to 1.69, −0.97 to 6.35, −1.35 to 1.59, −2.24 to 0.70, −0.77 to 5.63, −1.63 to 1.41, −2.38 to 0.40, and −1.36 to 2.15, respectively. The average Igeo values were in the order of Cd (3.28)> Hg (1.53) > Zn (0.55) > Cr (0.17) > Ni (–0.06)>As (–0.27)> Cu (–0.55)> Pb (–0.58). The average Igeo values of Cd and Hg were above 3 and accounted for 59% and 5% of the soil samples, respectively. These results indicate that this area was highly enriched and polluted with Cd and Hg. Cd and Hg had positive loadings in the PC1 and PC2 axes, and cluster 3 (samples affected by carbonate rocks) and cluster 4 (samples affected by human pollution) (Figure 4) were distributed around Cd and Hg, which indicates that the Cd and Hg in surface soil came from carbonate rocks (PC1) and human pollution (PC2). Carbonate rock in this study had high contents of Cd and Hg and carbon rock areas were more likely to accumulate PTEs than clastic areas [7]. At the same time, compared with deep soil, Cd and Hg are more concentrated in the surface soil [8]. Surface soil was collected in the study area, so the risk of contamination with Cd and Hg was higher in carbon rock area. In addition, the lime and cement factories increased the contents of Cd and Hg in the soil which had further aggravated the level of pollution (Figure 3 and Figure 6).

Due to the high pollution risk of Cd and Hg in this area, the study area may not be suitable for growing crops. However, rice planting areas were located in areas with high-levels of Cd, Hg, Zn, and Cr (Figure 6). We found that PTEs in rice grains, except for Cr and Cd, were at a safe level in the survey area according to the Chinese national food safety standard pollutant limits (GB2762 2017) (Table 1). The proportion of samples that exceeded the safety standards of China for pollutant limit for each metal followed this sequence: Cr (7.7%, *N* = 52) > Cd (3.85%, *N* = 52). The highest Cr and Cd concentrations were 1.92 and 1.52 times the limit values of 1 and 0.2 mg/kg, respectively. However, the Cd concentration in rice grains was not above the limit of 0.4 mg/kg from the Food and Agriculture Organization of the United Nations/World Health Organization (FAO/WHO) standards [28].

Areas exceeding the concentration limit of Cd in rice grains were located around the rock samples R24 and R20 and soil samples B128 (Cd 9.67mg/kg) and B136 (Cd 12.31 mg/kg) (Figure 1 and Figure 6). The sample with the highest Cd content in surface soil (B127 26.70 mg/kg) was also located in the excess Cd rice zone. High Cd concentrations obviously led to high contents in rice grains [16]. In addition, in the area located between the carbonate rock area and clastic rock area, the low pH also increased the Cd content in rice grains [42], while the pH situation was the opposite for Cr in rice grains. In natural, acidic systems, the reduction of soluble Cr (VI) to dissoluble Cr (III) occurs, decreasing the available Cr in soil [43]. The results show that rice grains with excess Cr were located in areas with high Cr content in the soil as well as areas with high pH, and Corg values (Figure 3 and Figure 6). Similar results were obtained in a previous study [43] in which high pH and Corg areas were associated with increased Cr contents in rice. So we speculated that areas with high Corg values and high Cd and Cr contents are not suitable for rice planting.

## 4. Conclusions

This research aimed to examine the PTE concentration and spatial variation and the soil–rice system quality in Babao town. We found that the average values of all PTEs were higher than the background values of Yunnan, and the surface soil was mainly threatened by Cd and Hg. The area southwest of Babao town (the carbonate rock area) had the highest PTE content, and this was related to the parent rock and human activities in this area. In the study area, 7 out of 17 carbonate rocks had Cd concentrations above 1 mg/kg, and carbonate area had high pH value and was riched with Corg, Mn, and TFe_2_O_3_. These good conditions led to the Cd concentrations of surface soil in the study area were as high as 26.70 mg/kg.

The lime plant and cement plant also caused increments in the PTE content in soil. Hence, these factories need to strengthen their supervision and management, especially in rural areas, where the factories are sometimes unsupervised.

In this area, only 2 out of 52 rice grains had Cd and 4 out of 52 rice grains had Cr concentrations slightly above the Chinese standards, which was related to the pH and Corg values. The areas with excess Cd of rice grains were located in low pH areas and the areas with excess Cr of rice grains were located in high pH and Corg areas. In addition to pH, Corg, Mn, and TFe_2_O_3_, other factors such as repeated rainfall in humid southern climates and irrigation water also affected PTE concentrations in soil and rice. However, these factors were not examined in this study. These factors should be further investigated in the study area.

## Figures and Tables

**Figure 1 ijerph-18-03122-f001:**
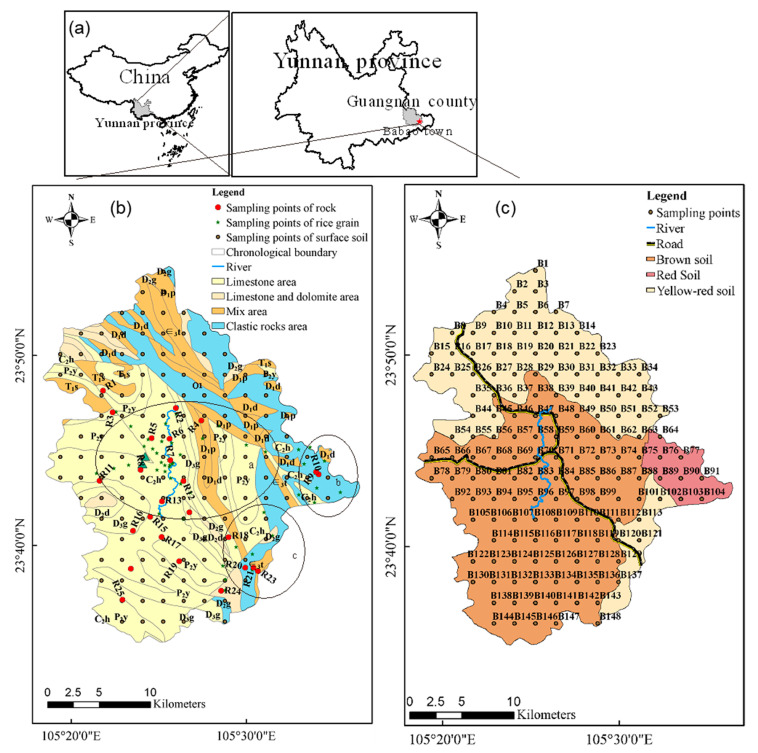
Location of the study area and distribution maps of sampling sites. (**a**) The core area of rice cultivation in Babao town; (**b**) Shadou rice cultivation area; (**c**) Baile rice cultivation area.

**Figure 2 ijerph-18-03122-f002:**
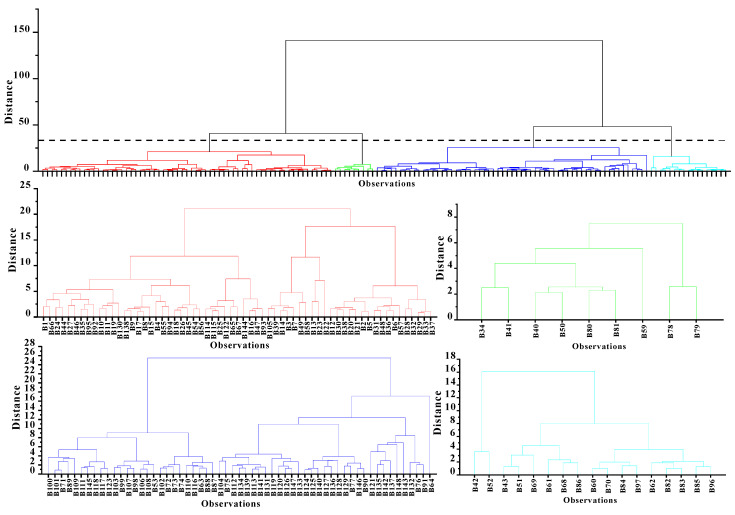
Dendrogram of the cluster analysis applied to all sampling sites. The dashed line represents the level of dissimilarity chosen to identify the four clusters selected for further analysis.

**Figure 3 ijerph-18-03122-f003:**
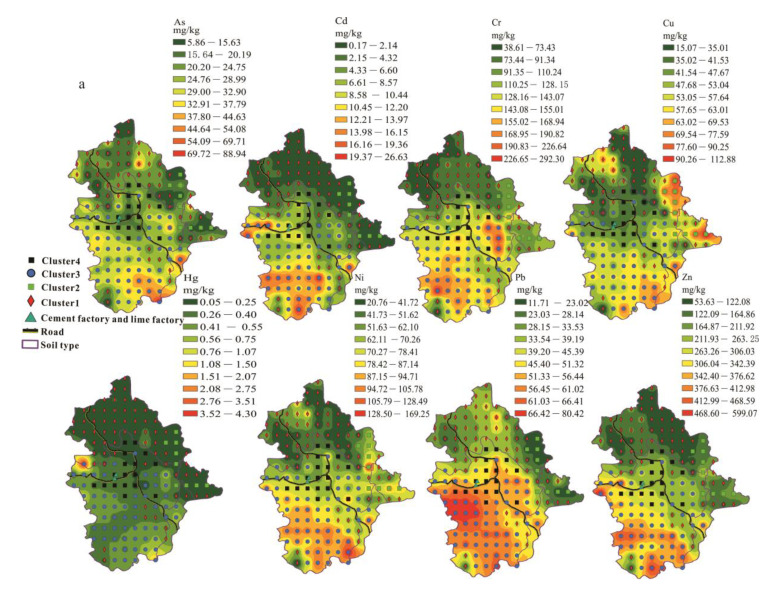
Concentration spatial distribution of PTEs and other elements. (**a**) Concentration spatial distribution of PTEs; (**b**) Concentration spatial distribution of other elements.

**Figure 4 ijerph-18-03122-f004:**
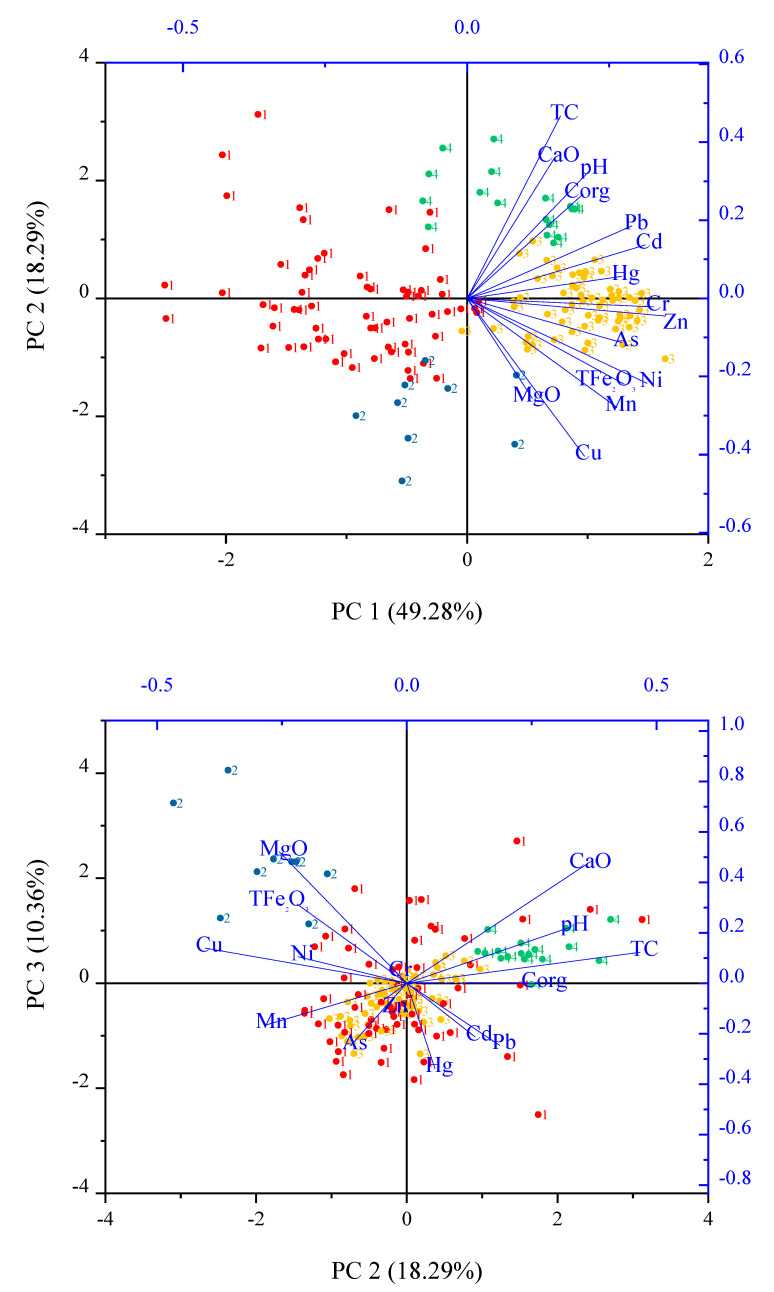
Biplots of the principal components for the clr-transformed whole composition of soil samples, with four groupings of soil samples. Cluster 1, cluster 2, cluster 3, and cluster 4 represent soil samples affected by both two types of rocks, clastic rock, carbonate rock, and human activity, respectively.

**Figure 5 ijerph-18-03122-f005:**
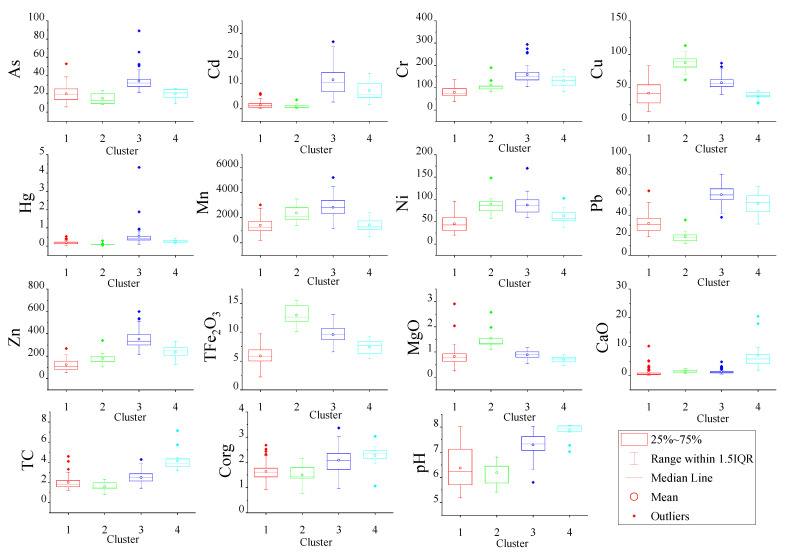
Box plots of soil classification for the elements.

**Figure 6 ijerph-18-03122-f006:**
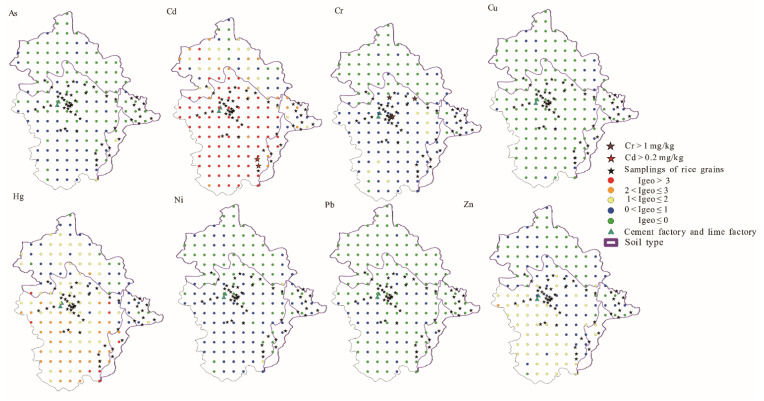
Geoaccumulation index (Igeo) showing the spatial distribution of PTEs in Babao Town.

**Table 1 ijerph-18-03122-t001:** Descriptive statistics of soil and grain samples in Babao Town. CV: variable coefficient; MLC: Maximum Contaminant Level. The units of TFe_2_O_3_, MgO, CaO, TC, Corg are percentages (%), and pH values are dimensionless. The units of other items are mg/kg.

Samples	Item	As	Cd	Cr	Cu	Hg	Ni	Pb	Zn
Surface soil	Mean	25.34	6.150	119.03	50.58	0.329	66.96	44.53	231.11
	SD	11.48	6.086	47.06	17.68	0.396	26.82	17.38	124.22
	CV/%	45.3	99.0	39.5	35.0	120.3	40.1	39.0	53.7
	Median	24.89	4.021	115.90	51.24	0.269	68.03	44.15	214.70
	Range	5.76–88.94	0.167–26.699	38.40–293.60	14.73–112.95	0.051–4.315	0.06–169.49	11.70–80.50	52.40–599.20
	Background of Yunnan [26]	18.40	0.218	65.20	46.30	0.058	42.50	40.60	89.70
Rice rain	Mean	0.12	0.043	0.40	1.64	0.004	0.31	0.067	17.92
	CV/%	22.6	124.2	119.8	41.6	45.9	60.8	68.1	18.5
	SD	0.19	0.440	0.03	0.68	0.52	0.001	0.02	0.02
	Median	0.12	0.026	0.11	1.64	0.003	0.36	0.06	18.79
	Range	0.04–0.19	0.009–0.303	0.07–1.92	ND–3.005	0.002–0.010	0.06–0.70	ND–0.281	10.94–24.09
	MLC(China ^a^)	-	0.2	1	10	0.02	-	0.2	50
	MLC(FAO/WHO)	0.2/0.35 ^b^	0.4 ^b^	-	-	-	-	0.2 ^b^	-
		Mn	TFe_2_O_3_	MgO	CaO	TC	Corg	pH	
Surface soil	Mean	2020.78	7.96	0.88	1.88	2.43	1.88	6.90	
	CV/%	48.0	31.3	38.5	145.6	37.9	25.8	12.1	
	SD	970.48	2.49	0.34	2.73	0.92	0.49	0.84	
	Median	1874.00	7.89	0.83	1.05	2.26	1.80	7.12	
	Range	204.16–5187.00	2.23–15.45	0.25–2.91	0.14–20.48	0.81–7.15	0.78–3.36	5.19–8.07	

^a^: Chinese national food safety standard pollutant limit (GB2762 2017) ^b^: FAO/WHO (2019) [28]. As: 0.2 mg/kg applies to polished rice, 0.35 mg/kg applies to husked rice; Cd: 0.4 mg/kg applies to polished rice; Pb: 0.2 mg/kg can apply to whole cereal grains not including buckwheat cañihua and quinoa.

**Table 2 ijerph-18-03122-t002:** Variance explained by each principal component and component loadings for PC1–PC3 of the soil samples.

Item	PC1	PC2	PC3
Percentage of Variance	49.28%	18.29%	10.36%
Cumulative Percent	49.28%	67.57%	77.93%
As	0.272	−0.113	−0.247
Cd	0.313	0.137	−0.213
Cr	0.328	−0.022	0.043
Cu	0.206	−0.404	0.139
Hg	0.271	0.056	−0.338
Ni	0.263	−0.279	−0.164
Pb	0.317	−0.218	0.105
Zn	0.29	0.186	−0.246
Mn	0.349	−0.045	−0.065
TFe_2_O_3_	0.276	−0.22	0.313
MgO	0.115	−0.255	0.521
CaO	0.154	0.359	0.471
TC	0.164	0.467	0.123
Corg	0.204	0.267	−0.001
pH	0.216	0.328	0.222

**Table 3 ijerph-18-03122-t003:** Soil/rock ratios of potentially toxic elements (PTEs) in carbonate rock and clastic rock.

Rock Type	Rock Number	Item	As	Cd	Cr	Cu	Hg	Ni	Pb	Zn
Carbonate rock	R2–R6, R9, R11–R19, R22, R23	Mean	0.55	0.69	10.55	1.68	0.014	13.15	2.65	4.21
		Median	0.31	0.40	10.60	1.71	0.014	13.30	2.14	3.85
		C_soil_/C_rock_	61.19	16.56	15.07	34.24	38.05	6.64	22.76	83.53
Clastic rock	R1, R7, R8, R10, R20, R21	Mean	4.86	0.70	53.02	26.42	0.043	30.95	17.51	71.08
		Median	4.68	0.14	51.65	26.70	0.027	31.90	15.55	72.05
		C_soil_/C_rock_	3.13	1.48	2.12	3.30	2.59	2.88	1.06	2.55

## Data Availability

The data presented in this study are available on request from the corresponding author. The data are not publicly available due to privacy.

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
