# Peer review of "Concentration and Spatial Distribution of Potentially Toxic Elements in Surface Soil of a Peak-Cluster Depression, Babao Town, Yunnan Province, China"

_ijerph, 2021, doi:10.3390/ijerph18063122_

Round 1

Reviewer 1 Report

The manuscript “Concentration and spatial distribution of heavy metals in surface soil of a peak-cluster depression, Babao town, Yunnan Province, China" provides a number of results that may be interesting for the scientific community.  However there are minor and major concern reported below. The manuscript should undergo moderate English editing, please address this during revision. Therefore, without this clarification, it is difficult for me to recommend the manuscript for publication in its present form in ijerph.

Heavy metals: it is best to use potential toxic metals (i.e. 10.1016/j.ecoenv.2017.11.041 ;  10.1007/s00128-019-02605-1 ; 10.1016/j.jenvman.2017.11.080)

Abstract "eight metal elements (As, Cd, Cr, Cu, Hg, Ni, Pb, and Zn) "
why not other elements?

Materials and Methods

"Figure 1" instead "figure"

"The annual average temperature is 17℃, and the frost
period is short. The average annual rainfall is 1056.5mm" which range of years were take into account?

Section 2.2 insert the standard methods and procedures

Why not was tested the bioavailability of the elements?

Results and discussion

Please use international and not just Chinese threshold as it is an international scientific research paper

"Figure
2Error! Reference source not found"  check it

What variance have the fourth axis of the PCA ?

An analysis of possible contaminations? the reasons? why so much cadmium?

Author Response

Response to Reviewer 1 Comments

We are very grateful for your comments regarding our manuscript 「Concentration and spatial distribution of potentially toxic elements in surface soil of a peak-cluster depression, Babao town, Yunnan Province, China. All your suggestions are very important to us, both for composing the manuscript and our further research. We have studied comments carefully and have made corrections.  We have carefully edited English language and the typographic errors in the revised manuscript which we hope meet with approval.

Point 1: Heavy metals: it is best to use potential toxic metals (i.e. 10.1016/j.ecoenv.2017.11.041; 10.1007/s00128-019-02605-1; 10.1016/j.jenvman.2017.11.080). 

Response 1: We are very grateful for your suggestion. We have changed the heavy metal to potential toxic metals in the paper. And thank you very much that you provide the reference. We read it and found them very useful. Thank you again for your additional valuable comments that helped us improve our manuscript.

Point 2: Abstract "eight metal elements (As, Cd, Cr, Cu, Hg, Ni, Pb, and Zn) " why not other elements?

Response 2: Many elements are relevant to the environment. Cr, Ni, Cu, Zn, Cd, Pb, Hg, and As were the most toxic heavy metals in the environment. And we add these texts to the part of the introduction. The abstract was also changed.

Point 3: "Figure 1" instead "figure".

Response 3: We are very grateful for your suggestion. We have formatted according to the requirements for authors and made the figures and tabs accordingly.

Point 4: "The annual average temperature is 17℃, and the frost period is short. The average annual rainfall is 1056.5mm" which range of years were take into account?

Response 4: We used these data came from a report and we do not know which range of years were take into account. So we search the data form website. And change it as ‘The annual average temperature is 16.7 °C(based on the Guangnan county from 1971-2000, http://www.weather.com.cn/cityintro/101290607.shtml), and the average annual rainfall is 1104.65 mm (based on the Funing station observations of 2001-2015)’.

Point 5: Section 2.2 insert the standard methods and procedures.

Response 5: We are very grateful for your suggestion.We insert the standard methods and procedures as follows: The following sampling and analysis methods were used: “Specification of Multi-Purpose Regional Geochemical Survey (1:250 000) (DZ/T 0258-2014)” and “Specification of Regional Biogeochemical Assessment (DZ/T 0289-2015)” issued by the China Geological Survey.

Point 6: Why not was tested the bioavailability of the elements?

Response 6: It is better to test the bioavailability of the elemensts. But we has limit fund that we have to give it up. And if we had extra money in the future, we will test the bioavailability of the elements.

Point 7: Please use international and not just Chinese threshold as it is an international scientific research paper.

Response 7: We are very grateful for your suggestion. We search the paper and internet and supplement the FAO/WHO standard in table 1. We found many article use old standard and we want to use newest standard, so we finally found the GENERAL STANDARD FOR CONTAMINANTS AND TOXINS IN FOOD AND FEED, CXS 193-1995, Adopted in 1995, Revised in 1997, 2006, 2008, 2009, Amended in 2010, 2012, 2013, 2014, 2015, 2016, 2017, 2018, 2019. But in this standard only regulated As,Cd,Pb content. Hg was regulated only in water and salt food grade. And we are very embarrassed that we searched about 10 hours and the results seemly not right.

Point 8: "Figure 2Error! Reference source not found" check it.

Response 8:  We are very grateful for your suggestion. I have changed it.

Point 9: What variance have the fourth axis of the PCA ?

Response 9: The varicance of PC4 was 5.98%, and the eigenvalue was 0.897.

Point 10: An analysis of possible contaminations? the reasons? why so much cadmium?

Response 10: We rewrite the Risk evaluation part and Enrichment and depletion of PTEs in soils part. The answer of question 1 and question 2 as follows: The average Igeo values were in the order of Cd (3.28)> Hg (1.53) > Zn (0.55) > Cr (0.17) > Ni (–0.06)>As (–0.27)> Cu (–0.55)> Pb (–0.58). The average Igeo values of Cd and Hg were above 3 and accounted for 59% and 5% of the soil samples, respectively. These results indicate that this area was highly enriched and polluted with Cd and Hg. Cd and Hg had positive loadings in the PC1 and PC2 axes, and cluster 3 (points represent carbon rock areas) and cluster 4 (points represent human pollution) (Figure 3) were distributed around Cd and Hg, which indicates that the Cd and Hg in surface soil came from carbon rock (PC1) and human pollution (PC2). Carbon rock had high contents of Cd and Hg and carbon rock areas were more likely to accumulated PTEs than clastic areas [7]. At the same time, compared with deep soil, Cd and Hg are more concentrated in the surface soil and are more easily affected by rainfall [8]. Surface soil was collected in the study area, so the risk of contamination with Cd and Hg was higher in carbon rock areas. In addition, the lime and cement factories increased the contents of Cd and Hg in the soil and aggravated the level of pollution (Figure 3, Figure 6).

 The answer of question 3 as follows: The Cdsoil/Cdrock in carbon rock was as high as 16.56, larger than ratios found in Lower Burgundy (ranged from 4.6 to 5.7) in France [41] and Huaxi of Guizhou Province (the largest ratio was 8.3) [25]. Carbon rock had a higher Cd content (mean value 0.69 mg/kg, median value 0.4 mgkg)( Table 2). Hence, rocks with high Cd concentrations cause the Cdsoil to be of a high level (Figure 4, Table 1), making Cd concentration as high as 26.70 mg/kg in the study area, much higher than that in other carbonate areas.

Reviewer 2 Report

Review:

Concentration and spatial distribution of heavy metals in surface soil of a peak-cluster depression, Babao town, Yunnan Province, China

The authors in their paper raised a crucial topic of heavy metals soil contamination. The analyzes included soil, rocks, and rice grains. The paper presents very interesting results obtained after statistical analysis and spatial data (ArcGIS). The presented research's scientific level is relatively high, while the authors should describe the submitted data in a more detailed manner.

Here are some minor notes:

1) The layout of the abstract is distorted. The authors discuss their research in one sentence, and the rest are the results. I propose constructing an abstract – introduction, research aim, methods used, and the most important results and conclusions.

2) 2nd sentence in the abstract: Cadmium, Hg, Zn ... is illogical - syntax error. Please consistently choose the full element names or their Cadmium symbol.

3) The Introduction lacks a clear research goal and research hypotheses. Please highlight it.

4) Fig. 2 is illegible - please correct this.

5) The text and figures should be formatted according to the requirements for authors. Please mark the figures and tabs accordingly.

Author Response

Response to Reviewer 2 Comments

Thank you very much for reading my article and for your many valuable suggestions.

Point 1: The layout of the abstract is distorted. The authors discuss their research in one sentence, and the rest are the results. I propose constructing an abstract – introduction, research aim, methods used, and the most important results and conclusions. 

Response 1: We are very grateful for your suggestion. Yes, the abstract is distorted and we did not notice.  We had constructed the abstract according to your suggestion. And the abstract as: Potentially toxic elements (PTEs) in Chinese agricultural soils, including those in some heritage protection zones, are serious and threaten food safety. Many scientists think that these PTEs may come from parent rock. Hence, at a karst rice-growing agricultural heritage area, Babao town, Guangnan County, Yunnan Province, China, the concentrations of eight PTEs (As, Cd, Cr, Cu, Hg, Ni, Pb, and Zn) were determined in 148 surface soil, 25 rock, and 52 rice grain samples. A principal component analysis (PCA) and hierarchical cluster analysis were used to divide the surface soil into groups, and inverse distance weighting (IDW) was used to analyze the spatial distribution of PTEs. Soil pollution was assessed with the geoaccumulation index (Igeo). The results show that Cd, Hg, Zn, and Cr were polluting the soil (average Igeo>0). The highest concentration of PTEs was distributed in the southwest of Babao town in the carbon rock area, which had the highest pH and soil total organic carbon (Corg), Mn, and TFe2O3 contents. PCA biplots of soil samples showed that the carbon rock area was associated with the most species of PTEs in the study area including Pb, Cd, Hg, As, and Zn. The clastic rock area was associated with Cu and Ni, and the lime and cement plants were associated with CaO, pH, Corg, TC, and aggravated PTE pollution around factories. In high-level PTE areas, rice was planted. Two out of 52 rice grain samples contained Cd and 4 out of 52 rice grain samples had Cr concentrations above the Chinese food safety standard pollutant limit (Cd 0.2 mg/kg; Cr 1mg/kg). Therefore, the PTEs from parent rocks are already threatening rice safety. The government should therefore plan rice cultivation areas accordingly.

Point 2: 2nd sentence in the abstract: Cadmium, Hg, Zn ... is illogical - syntax error. Please consistently choose the full element names or their Cadmium symbol.

Response 2: We are very grateful for your suggestion. We have changed it.

Point 3: The Introduction lacks a clear research goal and research hypotheses. Please highlight it.

Response 3: Thank you for your suggestion. It is a good suggestion. We have rewritten this part to highlight research goal and research hypotheses. The text as follow: Due to topographical limitations, agricultural production has become the main contributor to the local economy. Studies have shown that there is serious Cd and Hg pollution in the soil around this area [18,19]. However, little information is available on the quality of the overall soil–rice system in Babao town. Pollution may derive from parent materials and threaten the safety of the Babao production area. Therefore, we collected 148 surface soil samples, 52 rice grain samples, and 25 rock samples from the town to investigate the soil and rice situation. Cluster analysis was used to classify soil samples, and the results were combined with a geological statistical analysis to study the soil spatial variation of PTEs and the origins and associated risks of PTEs.

Point 4: Fig. 2 is illegible - please correct this.

Response 4: We are very grateful for your suggestion. We have changed the figure to make it clear.

Point 5: The text and figures should be formatted according to the requirements for authors. Please mark the figures and tabs accordingly.

Response 5: Thank you for your suggestion. I have changed text and figures formatted according to the requirements for authors and mark the figures and tabs accordingly.

Round 2

Reviewer 1 Report

The authors responded adequately to all comments. there are other minor issues that could be made.

Introduction

Cd (1.244 mg/kg):   1.2 instead 1.244 mg/kg)